# Problem Solving in Animals: Proposal for an Ontogenetic Perspective

**DOI:** 10.3390/ani11030866

**Published:** 2021-03-18

**Authors:** Misha K. Rowell, Neville Pillay, Tasmin L. Rymer

**Affiliations:** 1College of Science and Engineering, James Cook University, P. O. Box 6811, Cairns, Queensland 4870, Australia; tasmin.rymer@jcu.edu.au; 2Centre for Tropical Environmental and Sustainability Sciences, James Cook University, P. O. Box 6811, Cairns, Queensland 4870, Australia; 3School of Animal, Plant and Environmental Sciences, University of the Witwatersrand, Johannesburg 2000, South Africa; Neville.Pillay@wits.ac.za

**Keywords:** behavioural flexibility, cognition, development, individual, innovation, ontogeny

## Abstract

**Simple Summary:**

Animals must be able to solve problems to access food and avoid predators. Problem solving is not a complicated process, often relying only on animals exploring their surroundings, and being able to learn and remember information. However, not all species, populations, or even individuals, can solve problems, or can solve problems in the same way. Differences in problem-solving ability could be due to differences in how animals develop and grow, including differences in their genetics, hormones, age, and/or environmental conditions. Here, we consider how an animal’s problem-solving ability could be impacted by its development, and what future work needs to be done to understand the development of problem solving. We argue that, considering how many different factors are involved, focusing on individual animals, and individual variation, is the best way to study the development of problem solving.

**Abstract:**

Problem solving, the act of overcoming an obstacle to obtain an incentive, has been studied in a wide variety of taxa, and is often based on simple strategies such as trial-and-error learning, instead of higher-order cognitive processes, such as insight. There are large variations in problem solving abilities between species, populations and individuals, and this variation could arise due to differences in development, and other intrinsic (genetic, neuroendocrine and aging) and extrinsic (environmental) factors. However, experimental studies investigating the ontogeny of problem solving are lacking. Here, we provide a comprehensive review of problem solving from an ontogenetic perspective. The focus is to highlight aspects of problem solving that have been overlooked in the current literature, and highlight why developmental influences of problem-solving ability are particularly important avenues for future investigation. We argue that the ultimate outcome of solving a problem is underpinned by interacting cognitive, physiological and behavioural components, all of which are affected by ontogenetic factors. We emphasise that, due to the large number of confounding ontogenetic influences, an individual-centric approach is important for a full understanding of the development of problem solving.

## 1. Introduction

Increasing concerns over human-induced rapid environmental change has led to a corresponding increase in interest in understanding how animals will cope with these challenges. Rapid and unpredictable changes may have significant effects on survival and coping ability [1]. In order to survive, animals need to gain information about the environment (e.g., relative predation risk and food availability). While this might sometimes be easily attained, such as directly observing fruit on a tree, obtaining resources or avoiding predation may require an ability to solve a problem, such as obtaining fruit that is out of reach.

Problem solving has been documented in all major vertebrate taxa, including mammals (e.g., food-baited puzzles in various mammalian carnivores, [2]), birds (e.g., food-baited puzzles given to multiple parrot and corvid species [3,4]), reptiles (e.g., multiple species of monitor lizards *Varanus* spp. are capable of solving food-baited puzzle boxes, [5]), amphibians (e.g., detour task, where the animal had to move around an obstacle in brilliant-thighed poison frogs *Allobates femoralis*, [6]), fishes (e.g., foraging innovation in guppies *Poecilia reticulata*, [7]), and some invertebrates (e.g., overcoming a physical barrier in leaf-cutting ants *Atta colombica* [8]).

Currently, there is no universally accepted definition of problem solving (Table 1). From our literature search (see below), most definitions consider mechanical (i.e., movements required to solve problems), morphological (i.e., physical structure to manipulate objects to solve a problem) and/or cognitive (i.e., assessing, learning, storing information about problem) components as part of problem-solving ability. We consider problem solving to be the ability of an individual to integrate the information it has gained (knowledge or behaviour) to move itself, or manipulate an object, to overcome a barrier, negative state or agent, and access a desired goal or incentive, such as a resource [9,10]. Most reports of problem solving are based on experimental evidence where animals are presented with a feeding motivation task (e.g., a puzzle box or detour task), in which an animal manipulates an object, or moves itself around the object, to access the food. Occasionally, animals are experimentally presented with an obstacle blocking access to a location, and the animal needs to move the obstacle to access a refuge or their nest. These solutions can be achieved by innovation (the use of a new behaviour, or existing behaviour in a new context [11]) and/or by refining behaviour over repeated sessions with the stimulus (e.g., trial-and-error learning). Our literature search has also demonstrated that problem solving is sometimes assessed simply as a dichotomous skill, in which an animal either can or cannot solve a problem, but other studies have focused on how animals vary in the way they solve problems, and how efficiently they solve problems. Our definition encompasses all of these aspects. 

Successful problem solving has been theorised to be important for survival, as it allows animals to adjust to changing environmental conditions [24] and even invade new environments (e.g., bird species introduced to New Zealand, [25]), or to cope with harsh or extreme conditions [26]. However, the ability of animals to solve problems [27], and the specific strategy/manoeuvre that they use to solve problems [28], is highly variable, and this variation can be observed at all taxonomic levels, including between families (e.g., Columbida vs. Icteridae, [29]), genera (e.g., *Molothrus* vs. *Quiscalus* [30]), and species (jaguar *Panthera onca* vs. Amur tiger *P. tigris*, [2]). It is even possible that problem solving is phylogenetically conserved, with some groups having a greater potential to solve problems than others [31]. However, variation in problem-solving ability also occurs within species, including between populations (e.g., house finches *Haemorhous mexicanus* given extractive foraging tasks [32]), and individuals (e.g., meerkats *Suricata suricatta* given food-baited puzzle boxes [27]). Likely causes of this variation are the conditions that arise during an individual’s development. This variation could then allow problem-solving ability to be acted upon by natural selection [33], possibly impacting individual fitness. Therefore, understanding the influence of developmental factors on problem-solving ability is important. 

An individual’s behaviour, physiology and morphology may change as it grows and ages due to developmental changes in life history traits [34,35]. Furthermore, interactions and experiences with other individuals and the immediate environment further feedback into these systems [36]. These intrinsic and extrinsic factors, either independently or synergistically, influence the individual’s ability to cope with, and respond to, environmental challenges [37], although their outcomes are likely difficult to predict because of myriad interacting factors.

Although aspects of behaviour, physiology and cognition have been studied in an ontogenetic context [38,39], little is currently known about how problem-solving abilities develop and change as individuals grow and age. Developmental differences between individuals could fine tune or modulate the ability to solve problems, causing individual variation in this ability. Importantly, this inter-individual variation in problem solving could have fitness consequences by influencing survival and/or reproductive success. However, untangling the relative influence of intrinsic (genetic, neuroendocrine and aging) and extrinsic (environmental) factors on the development of problem solving is challenging [40,41]. We propose that an integrated approach, focusing on the development of problem solving, is needed to fully appreciate the ability and propensity of animals to solve novel problems. Our aim was to review the literature on problem solving to document and then construct the links between intrinsic and extrinsic factors that influence the development of problem-solving.

We therefore conducted a literature search using Google Scholar and the Web of Science database. We included the general search terms “problem solving” “innovation” and “animal” in all searches and excluded all articles with the word “human”. This produced 6100 hits. We further refined the search by including the following as specific terms in individual searches: “development”, “ontogeny”, “heritability”, “personality”, “cognition”, “learning”, “experience”, “age”, “hormone”, “brain”, and “environment”. Articles that were repeated in subsequent searches were ignored. Articles were excluded if: (1) the researchers trained the animals to solve the problem before testing (and, therefore, tested memory rather than natural problem-solving ability); (2) the authors referred to a type of problem solving that did not meet our definition (e.g., relational problems where animals needed to extract and transfer rules between tests); and/or (3) development of problem solving was not investigated. If two papers found similar results (e.g., neophobia hinders problem solving in a bird species), we only reported on one study to avoid repetition and to reduce the overall number of citations. 

Numerous studies have shown that animals can problem solve [42], and several studies have explored the fitness consequences of problem solving in animals (e.g., [10]). However, how problem solving develops is an area that has been little explored. In this paper, we first discuss how intrinsic and extrinsic factors influencing the ontogeny of individuals could affect the development of problem-solving ability. We focus on genetic (direct and indirect), neuroendocrine, and environmental (physical and social) factors, as well as age, learning and experience. Given the relative paucity of empirical studies investigating the development of problem solving in general (42 publications found of seven developmental factors), we demonstrate first how these factors impact other traits in order to create a conceptual framework for addressing problem solving. We acknowledge that limited information currently makes it challenging to separate developmental factors underlying problem-solving ability from other causal mechanisms (e.g., hormones, genetic effects). We then explore how interactions between intrinsic and extrinsic factors during an individual’s development could influence problem solving indirectly. Specifically, we focus on how personality (individual differences in behaviour) and behavioural flexibility (ability to change behaviour in response to environmental cues) contribute to differences in problem-solving ability. Finally, we briefly discuss aspects that have been overlooked in studies investigating the development of problem solving, providing hypotheses for future testing. Throughout this paper, we advocate for an individual-centric approach to study the ontogeny of problem solving, where individual variation in solving ability is considered, rather than only using simple population-level averages. Future studies should be tailored to focus on individual differences within and between tests, as well as consider a longitudinal approach to track how individuals change over their lifetimes. Analyses of these experiments should then include individual data points as a measure of individual ability and variation, and should not exclude outliers because these account for the species- or population-level variation.

## 2. Factors Affecting the Development of Problem Solving

Problem solving is influenced by direct [43] and indirect (epigenetic and transgenerational) genetic [44], and neuroendocrine [45] factors (Figure 1). Furthermore, extrinsic factors, including both the physical and social environments, can also affect the development of problem solving (Figure 1). However, the development, and ultimately expression, of problem solving is more likely impacted by complex interactions between these intrinsic and extrinsic factors (Figure 1), and is also likely to change as the animal ages and experiences (i.e., learns) new situations (e.g., ravens *Corvus corax* [28]; North Island robins *Petroica longipes*, [46]). Untangling these effects is likely to be challenging.

### 2.1. Instrinic Factors

#### 2.1.1. Direct Genetic Effects

Heritable genetic effects influence the development of phenotypic traits. For example, physiological stress (barn swallows *Hirudo rustica*, [47]), parental care (African striped mice *Rhabdomys pumilio* [48]), exploratory behaviour (great tits *Parus major* [49]), multiple aspects of cognition in chimpanzees *Pan troglodytes* [50], learning in hens *Gallus gallus domesticus* [51] and spatial learning ability (C57BL/6Ibg and DBA/2Ibg mice *Mus musculus* [52]) all have a heritable component (but see [53]). 

Heritable genetic effects may also affect the development of problem solving (Figure 1), although this has received little attention in the literature. Elliot and Scott [43] found that different dog *Canis lupus familiaris* breeds solved a complex barrier problem in different ways, and Audet et al. [54] showed that an innovative species of Darwin’s finches *Loxigilla barbadensis* had higher glutamate receptor expression (correlated with synaptic plasticity) than a closely related, poorly innovative species *Tiaris bicolor*. Tolman [55] and Heron [56] also indicated underlying genetic effects on maze-learning ability in rats, although the ability to learn a maze may not necessarily imply an ability to solve a problem (see [57]). In contrast, Quinn et al. [58] and Bókony et al. [59] found little measurable heritability of innovative problem-solving performance in great tits in a food-baited puzzle box and an obstacle-removal task, respectively. These studies suggest that the genetic architecture underlying problem solving may provide a rich area for future research.

#### 2.1.2. Indirect Genetic Effects

Indirect genetic factors, specifically epigenetic and transgenerational effects, influence how genes are read (e.g., DNA methylation, [60]) or expressed (e.g., hormones activating genes during sexual maturation, [61]) without altering the underlying DNA. These epigenetic changes are underpinned by biochemical mechanisms that affect how easily the DNA can be transcribed [62], subsequently influencing the development of different systems. For example, the activation of thyroid receptor genes (TRα and β) in the cerebellum of 0–19 day old chicks causes hormone-dependent neuron growth and development [63]. No studies to date have explored the effects of epigenetic factors on the development of problem solving, although this relationship can be postulated (Figure 1), since epigenetic factors influence the development of behaviour (e.g., maternal care, [64]), and cognition (e.g., memory, [44]). Memory is an important component of problem solving [65]. Consequently, two possible routes could be inhibited via transcriptional silencing of the memory suppressor gene protein phosphatase 1 (PP1), and demethylation and transcriptional activation of the synaptic plasticity gene reelin, both of which enhance long-term potentiation. These could lead to increased memory formation (e.g., in male Sprague Dawley rats *Rattus norvegicus domesticus*, [44]).

Transgenerational epigenetic effects can also influence development. These effects result from parental or grandparental responses to prevailing environmental conditions, which influence how offspring and grand offspring ultimately respond to their own environment [66]. For example, embryonic exposure to the endocrine disruptor vinclozilin in female Sprague Dawley rats resulted in epigenetic reprogramming of hippocampal and amygdala genes for at least three generations, with the resulting F3 males showing decreased, and F3 females showing increased, anxiety-like behaviour, as adults [67]. An interesting avenue for research into transgenerational effects on the development of problem solving is the NMDA (N-methyl-D-aspartate) receptor/cAMP (cyclic adenosine monophosphate)/p38 MAP kinase (P38 mitogen-activated protein kinases) signalling cascade. Exposure of newly weaned Ras-GRF1 (growth regulating factor) knockout mice to an enriched environment enables this latent signalling pathway, rescuing defective long-term potentiation and learning ability [68]. These epigenetic effects may therefore influence problem-solving ability indirectly by affecting the individual’s learning ability, or possibly directly by affecting the development of particular brain regions.

#### 2.1.3. Neuroendocrine Effects—Brain Morphology

Many developmental processes are driven by neuroendocrine factors that are, themselves, impacted by other developmental processes [63]. While the development of many of the brain’s circuits (e.g., those located near the sensory or motor periphery), are governed by innate mechanisms [69], other parts (e.g., the basolateral nucleus of the amygdala and the cerebellar cortex [70]; the CA1 region of the mammalian hippocampus [71]; the avian hippocampus [72]) are considerably more plastic and more responsive to external stimuli, maintaining a high degree of neural plasticity throughout life. As these brain regions can be important for the expression of particular behaviours (e.g., the cerebellum is necessary for tool use, [73]), this plasticity has particular relevance for problem solving. For example, North American bird species with relatively larger forebrains were more likely to innovate when foraging than bird species with smaller forebrains [74] and New Caledonian crows *Corvus moneduloides*, which are renowned for their tool use and problem-solving abilities, had relatively larger brains than other bird species [75]. Similarly, C57BL/6J laboratory mice that received lesions to the hippocampus and medial prefrontal cortex initially showed impairments in solving a puzzle box task, although the mice ultimately solved the task over time, indicating the importance of experience and learning with repeated presentation of the task [76].

#### 2.1.4. Neuroendocrine Effects—Hormones

The brain is also the central control of endocrine responses that can influence an individual’s development (Figure 1). For example, the hypothalamic-pituitary-gonadal (HPG) axis activates gonadotropin-releasing hormone (GnRH), which stimulates the pituitary to produce luteinizing hormone (LH) and follicle-stimulating hormone (FSH, [77]). These hormones regulate the production of steroid hormones (testosterone and oestrogen) via the gonads [78], stimulating sexual maturity [79]. Fluctuations in steroids also influence cognitive function [80,81]. For example, female rats injected neonatally with testosterone show heightened learning of a Lashley III maze (contains start box, maze, and goal box; used to test learning and memory) as adults compared to non-injected females, although the underlying impacts on neural development or neuroendocrine processes were not discussed [82]. 

Endocrine responses can also feedback to brain morphology (Figure 1), affecting neural structure and function, which can impact behaviour, cognition, and development. The hypothalamic-pituitary-adrenocortical (HPA) axis regulates the secretion of adrenocorticotropic hormone (ACTH), which in turn regulates the secretion of glucocorticoid stress hormones (e.g., corticosterone, [83]) from the adrenal glands [84]. Short-term exposure to corticosterone can improve learning, since it allows important associations to be formed, such as between threat and a behavioural response [85]. However, prolonged increased corticosterone concentrations (chronic physiological stress) reduce hippocampal neuron survival [86], which interferes with learning [87,88], memory retrieval [89] and problem solving. For example, house sparrows *Passer domesticus* with prolonged elevated corticosterone concentrations were less efficient problem solvers of puzzle boxes than birds with lower corticosterone concentrations, as stress impairs working memory and cognitive capacity [45]. Prolonged physiological stress can also cause detrimental developmental changes in morphology (e.g., chickens [90,91]) and behaviour (e.g., rats [83]).

In contrast to stress hormones, the mesolimbic dopaminergic system [92], which consists of the substantia nigra and ventral tegmental region [93], regulates the production of dopamine, a hormone associated with motivation and reward-seeking [94]. Motivation is a physiological process [94] that increases persistence and thereby increases the likelihood of successfully solving a problem [95]. Persistence is important for problem solving in foraging tasks in house sparrows [96], common pheasants *Phasianus spp*. [97] and Indian mynas *Acridotheres tristis* [98], and in puzzle box tasks in spotted hyenas *Crocuta crocuta* and lions *P. leo* [99]. Changes to dopamine production can also negatively impact the development of sensorimotor integration [100], disrupting approach, seeking and investigatory behaviours [101] and acquisition of spatial discrimination [102]. Disruption to dopamine production, or other circuits, may also lead to an individual persisting with an inadequate strategy if the individual lacks inhibitory control [103] and cannot recognise when to terminate the behaviour [104]. Disruptions to these behaviours and cognitive functioning therefore impact foraging and exploratory behaviours [87,104], which can lead to undernutrition, and consequent negative impacts on growth and physical, behavioural, and cognitive development [105].

Other hormones have also been implicated in the expression of problem solving. For example, both norepinephrine and serotonin likely impact problem solving, since they are related to cognitive flexibility (e.g., rhesus macaques *Macaca mulatta* [106,107]), with serotonin activating, and norepinephrine deactivating, the prefrontal cortex [108]. However, although some studies have investigated the role of these hormones in problem solving, these relationships are not clearly defined. For example, dietary deficiency in n-3 fatty acids during development increased serotonin receptor density and reduced dopamine receptor binding in the frontal cortex of rats, and it also altered dopamine metabolism [109,110]. This dietary n-3 fatty acid deficiency also impaired problem solving in a delayed matching-to-place task in the Morris water maze [111]. However, whether problem-solving ability was impacted specifically by down-regulation of dopamine receptor binding, or up-regulation of serotonin receptor binding, is unclear.

### 2.2. Extrinsic Factors

#### 2.2.1. Physical Environmental Factors

The physical environment varies in structural complexity and quality across both spatial and temporal scales [112]. Throughout its lifetime, an individual will experience daily and/or seasonal variation in environmental conditions (e.g., rainfall, temperature, food availability, [113]), and/or when it disperses [114], migrates [115] or travels into different areas. This variability changes the likelihood of an individual encountering positive (e.g., food [116]) or negative (e.g., predator [117]) stimuli, consequently influencing its development (Figure 1). For example, a higher density and abundance of aquatic snails results in the development of larger pharyngeal jaw muscles and stronger bones in predatory pumpkinseed sunfish *Lepomis gibbosus* [111]. 

Some studies have investigated the interplay between physical environmental conditions and problem-solving ability. Favourable environmental conditions can reduce stress [118], promote active and exploratory behaviours [119] and enhance cognition [120], but harsh conditions may promote problem solving. For example, mountain chickadees *Poecile gambeli* living in harsher high elevation montane habitats with longer winters solved novel foraging problems significantly faster than chickadees living at lower elevations, most likely because finding food in these habitats was more challenging, and survival depends on plastic responses to these challenges [26]. However, this effect on food-motivated problem-solving ability was not seen in great tits experiencing similar harsh conditions [40], suggesting that species-dependent developmental factors may be constrained by environmental effects. Urban environments may also promote the development of problem solving since they are expected to contain a higher frequency of novel problems for animals to solve. For example, house sparrows [121] and house finches [32] in urban environments were more adept food-motivated problem solvers than birds from rural areas, particularly when the problem was difficult to solve [96].

#### 2.2.2. Social Environmental Factors

The social environment also changes throughout an individual’s lifetime, and has the capacity to influence its development (Figure 1). Any positive (e.g., offspring suckling from mothers) or negative interactions (e.g., siblings fighting over food) between individuals can be considered social, and can vary over time scales (e.g., from daily interactions between individuals in a group, to shorter interactions between parents and offspring or mating partners [122]). 

For mammals, females are constrained to care for their offspring through pregnancy and suckling [123]. Consequently, the mother’s physiological state and access to resources can impact offspring embryonic development prenatally through direct transfer of maternal hormones or nutrients across the placenta [124]. For example, pregnant female Sprague Dawley rats exposed to unpredictable, variable stress (e.g., restraint, food restriction) during the final week of gestation produced anxious daughters and sons with impaired cognitive function (contextual memory [125]). Furthermore, maternal care during postnatal development [64], particularly the mother’s diet quality, can also influence development. For example, protein deficiency in African striped mouse *Rhabdomys dilectus chakae* mothers during early postnatal development of offspring resulted in these offspring showing increased anxiety, decreased novel object recognition and increased aggression as adults compared to mice raised by mothers that did not experience nutrient deficiency [126]. Thus, detrimental developmental effects such as these may go on to impede offspring problem solving abilities.

For some species, a key developmental milestone is dispersal. Interactions with other conspecifics during this phase are often driven by dramatic developmental changes often associated with reproduction [114]. For example, male vervet monkeys *Chlorocebus pygerythrus* leave their natal group at sexual maturity and attempt to attain dominance in another group [127], which could lead to increased access to food resources that can be channeled further into growth and development. This process of leaving the natal territory, and any social interactions during this time, can feedback to the individual to further affect its development. For example, in many species (e.g., brown rats), dispersing juveniles undergo a period of heightened exploration and learning, allowing them to rapidly adjust to new environmental conditions [128]. However, it is unknown how dispersal and other associated events impact an individual’s problem-solving abilities.

Problem solving is most often studied in social animals [122], possibly because they are more conspicuous than solitary species. In some species, such as European starlings *Sturnus vulgaris* with a foraging task [129], coyotes *Canis latrans* with a puzzle box task [130] and rhesus macaques in an associative learning task [131], dominant individuals are better learners and problem solvers. Similarly, the presence of an alpha individual impedes problem solving success in subordinate spotted hyenas presented with a puzzle box [132] and ravens in a string-pulling task [28] due to direct interference and increased aggression from the dominant. However, in other species, such as blue tits *Cyanistes caeruleus* [133], adult meerkats [27] and chimpanzees [134], subdominants tend to be better solvers of puzzle boxes, since their lower competitive ability makes them more reliant on alternative methods for accessing resources [26]. Group size may also influence problem solving, although results are equivocal. For example, larger groups of house sparrows [121] and Australian magpies *Gymnorhina tibicen* [135] in extractive foraging tasks and zebra fish *Danio rerio* in an avoidance task [136] were better problem solvers than individuals in small groups, possibly because larger groups contained more reliable demonstrators. However, orange-winged amazons *Amazona amazonica* had similar solving success in a string-pulling task when tested in groups or in isolation [137]. Social carnivore species, such as banded mongoose *Mungos mungo*, were also less successful problem solvers of a puzzle box compared to solitary species, such as black bears *Ursus americanus* and wolverines *Gulo gulo*, suggesting that relative brain size may be more important for cognitive abilities than social environment [33].

Problem solving studies in solitary species are generally lacking, making it difficult to assess how social interactions may impact the development of problem solving in these species. However, it is evident that individual animals can solve problems in the absence of conspecifics. For example, black-throated monitor lizards *V. albigularis albigularis* [138], eastern grey squirrels *Sciurus carolinensis* [139], and orangutans *Pongo pygmaeus* [140] can individually solve puzzle boxes using flexible behaviours (i.e., switching strategies when necessary), persistence and learning. Similarly, North Island robins [46] and brilliant-thighed poison frogs [8] can solve detour problem tasks when tested in their home territories. How solitary species solve problems in the presence of conspecifics, however, is an area for future investigation.

## 3. Interacting Factors that Influence the Development of Problem Solving

### 3.1. Gene × Environment Interactions

Genotype × environment interactions can also have a profound effect on the development of individuals (Figure 1). For example, the gene monoamine oxidase A (*MAOA*) encodes for an enzyme that impacts serotonergic activity in the central nervous system, leading to increased impulsivity and anxiety [141]. Stressful life events, or changes in social structure or status can alter the expression of this gene, leading to developmental changes during adulthood. For example, rhesus macaques raised in the absence of their parents showed increased aggression due to low *MAOA* enzymatic activity [142]. 

Although genotype × physical environment interactions have not been explored in the context of problem solving, environmental enrichment in captive bi-transgenic CK-p25 Tg laboratory mice is associated with the activation of plasticity genes, inducing chromatin modification via histone acetylation and methylation of histones 3 and 4 in the hippocampus and cortex, leading to increased numbers of dendrites and synapses [143]. This cascade of genetic and neuroendocrine processes functions to help restore learning and memory [143], both of which are important for problem solving [65,95]. 

Parents may also alter the environment (e.g., amount of parental care or food) their offspring experience [66], which could be a consequence of genetic variation between mothers [144] or a result of other factors (e.g., variability in resource availability [145]). When an offspring’s development is impacted by this nongenetic parental environment, these effects are known as parental effects [146], which are specific types of indirect genetic effects (IGEs, [144]). For example, female Long-Evans hooded rats that provided high levels of tactile stimulation (e.g., grooming and nursing [64]) to their young produced daughters that also displayed higher levels of maternal care to their own offspring [147], indicating an IGE. 

Maternal care also regulates the expression of the hippocampal glucocorticoid receptor gene by changing the acetylation of histones H3-K9 and the methylation of the NGFI-A consensus sequence on the exon 17 promoter [148]. Young rats that experienced low levels of maternal tactile stimulation showed reductions in hippocampal neuron survival [149] and decreased hippocampal glucocorticoid receptor mRNA expression [148], leading to chronic corticosterone release as adults [150]. Offspring also showed decreased exploratory behaviour [151] and impairments in spatial learning and memory [64] and object recognition [149,152] as adults. As for genotype × physical environment interactions, how the social environment × genotype interaction affects problem solving is a promising avenue for future research.

### 3.2. Neuroendocrine × Environment Interactions

Habitat complexity, resource availability and social complexity can influence development via effects on neuroendocrine systems, which can also result in changes to the social environment that may then feedback to further impact development. For example, nine-spined sticklebacks *Pungitius pungitius* preferentially shoal together in marine environments with high predation risk and patchy food resources, but prefer to swim alone when these constraints are relaxed in freshwater ponds [153]. Marine fish with more social interactions had significantly larger olfactory bulbs and optic tecta, parts of the brain associated with sensory perception, compared to solitary fish from freshwater ponds that experienced fewer social interactions [154,155]. Rhesus macaques from larger social groups also had more grey matter and greater neural activity in the mid-superior temporal sulcus and rostral prefrontal cortex than macaques from smaller groups [156]. Similarly, structurally complex, changing environments improve survival of hippocampal cells and neurons by increasing the level of nerve growth factor in the hippocampus [112], which increases hippocampal volume [83], leading to increased neural plasticity [157] and a greater capacity to adjust to new environmental conditions [158]. Environmental enrichment has also been shown to enhance long-term potentiation in the hippocampus, which facilitates learning and memory [159], two important processes for problem solving [23,95]. Environmental enrichment has been associated with increased problem-solving ability in C57/BL6J mice in an obstruction puzzle task [160] and Labrador retrievers in puzzle box tasks [161]. This suggests causal links between the environment, the neuroendocrine system, and problem solving which are likely mediated by underlying genotype × environment interactions. 

### 3.3. Age Effects

Separating out the effects of aging and neuroendocrine or genetic effects on development is challenging. Nevertheless, age-specific effects on development, regardless of the underlying mechanisms, are an important consideration. 

The nervous system shows age-dependent decreases in neurogenesis and plasticity, particularly in the dentate gyrus of the hippocampus [162], and the subventricular zone of the lateral ventricle [163], and these age-dependent changes can alter cognitive ability and behaviour (e.g., beagles [164]). Other neuroendocrine processes also naturally change with age. For example, as brown rats age, the ACTH response increases, glucocorticoid receptor binding capacity in the hippocampus and hypothalamus decreases, corticotropin releasing hormone (CRH) mRNA expression decreases in the paraventricular nucleus, and mineralocorticoid mRNA expression in the dentate gyrus of the hippocampus is reduced [165]. These changes result in an associated attenuation of the corticosterone response to novelty [164], as well as declines in spatial learning and memory [166]. 

Depending on the age of the individual, changes to both the physical and social environments also impact development [167]. When raised in small cages with limited space, juvenile rats showed increased anxiety, and lower activity and exploration, whereas older rats did not [167]. Similarly, older rats reared in larger groups were more active than juveniles, mostly likely due to increased frequency of social interactions and establishment of their rank within the social hierarchy [167].

Several studies have shown that juveniles are better problem solvers than adults, although the underlying mechanisms are currently not known. For example, juvenile Chimango caracaras *Milvago chimango* were more successful at solving a puzzle box task than adults [168], and juvenile canaries *Serinus canaria* solved a vertical-string pulling task, whereas adults did not [169]. Similarly, juvenile Chacma baboons *Papio ursinus* solved a hidden food task more often than adults [170], and juvenile kakas *N. meridionalis* showed higher innovation efficiency than adults across different tasks and contexts [171]. Juveniles are often prone to higher levels of exploration [159], and are more playful [172], than adult animals, allowing juveniles to rapidly gain motor skills [172]. This could possibly improve problem solving abilities of juveniles despite their lack of experience at solving tasks. However, results are species-specific, as Indian mynas [173] and spotted hyenas [174] show no age-specific effects on problem solving in foraging tasks, while adult meerkats [27] and black-capped chickadees [175] were better innovators than juveniles in extractive foraging tasks.

### 3.4. Learning and Experience

As an animal ages, it encounters predators and food resources, and interacts with conspecifics. These experiences provide a rich potential for learning, which is a critical component of problem solving. However, separating out the effects of the experience itself on development from other extrinsic and intrinsic factors, or their interactions, is challenging. Nevertheless, as in aging, an animal’s development can be impacted by its experiences, particularly via learning, suggesting that experience must be considered when attempting to understand how problem solving develops. 

To survive, use new resources, or avoid predators, individuals must learn to associate the experience with its significance (e.g., threat of a predator [176,177]). Learning enables animals to acquire information about the state of their environment [178] and learning through experience allows for adjustments in physiological and behavioural responses [176]. For example, repeated foot shock in a specific environmental location caused increases in norepinephrine and epinephrine in Sprague Dawley rats, eliciting fear and resulting in rats avoiding that location [179]. Similarly, guppies decreased their time foraging in the presence of a predatory convict cichlid *Cichlasoma nigrofasciatum* [180]. Animals can learn to solve problems in different ways, such as through trial and error (e.g., rooks *C. frugilegus* across multiple foraging extraction tasks [181]) or socially through local enhancement (e.g., common marmosets *Callithrix jacchus* in a foraging extraction task [182]), social facilitation (e.g., capuchin monkeys *Cebus apella* in a foraging extraction task [183]) or copying/imitation (e.g., laboratory rats in an extractive foraging task [184]). Learning from previous experience is also an important component for successful problem solving. For example, grey squirrels improve their ability to solve a food-baited puzzle box with repeated exposures to the problem [23]. Similarly, North Island Robins became more efficient problem solvers of new food-extraction tasks with experience [46]. 

### 3.5. Behavioural Flexibility and Personality

Although development is governed by several unifying genetic and physiological mechanisms, and these processes are impacted by age and environmental effects [185], the development of one individual differs considerably from that of another individual. Some of this variation can be attributed to the behavioural flexibility of each individual [29] and/or its personality [168], which also undergo developmental changes over the course of an individual’s lifetime [36]. 

Behavioural flexibility is the ability to switch behavioural responses (likely due to cognitive flexibility [95]) to adjust to new situations or states [186], and is likely governed by both genetic and non-genetic mechanisms [187]. The degree of behavioural and cognitive flexibility, and corresponding learning ability, is important for problem solving, as seen in tropical anoles (*Anolis evermanni* in an obstruction task [188]; *A. sagrei* in a detour task [189]), spotted hyenas in a puzzle box task [174], grey squirrels in a food-extraction task [139] and keas *Nestor notabilis* in a foraging extraction task [190]. However, the degree of flexibility varies between species. For example, Indian mynas are more flexible, and are better innovative foraging problem solvers, than noisy miners *Manorina melanocephala* across a range of tasks [173]. Importantly, individual differences in behavioural and cognitive flexibility, particularly learning ability, are often attributed to physiological effects occurring during development (e.g., corticosterone exposure in nestling Florida scrub jays *Aphelocoma coerulescens* [191]). 

An individual’s development and experiences can also affect its personality [192], defined as consistent individual differences in behaviour shown across contexts and situations, and over time [193]. Personalities are often measured along different axes (e.g., bold/shy [194]; proactive/reactive [195]), and are mediated by hormones [196]. Although personality itself is influenced by intrinsic (e.g., hunger [197]) and extrinsic (e.g., environmental quality [119]) developmental factors, personality can further feedback on an individual’s development through its effects on exploration [167]. For example, avoidant individuals may be less willing to investigate their environment than exploratory individuals, which reduces their chances of being predated, but also reduces foraging rate, which affects growth, as seen in grey treefrog tadpoles *Hyla versicolor* [198]. 

Personality can also impact problem solving [40]. Exploratory individuals have higher interaction rates with problems, increasing their likelihood of solving innovative tasks. For example, brushtail possums *Trichosurus vulpecula* that were exploratory, active and vigilant were more likely to solve an escape-box task during the first trial, and were capable of solving a difficult task, compared to less exploratory, less active and less vigilant individuals [199]. Similarly, exploratory fawn-footed mosaic-tailed rats *Melomys cervinipes* were faster problem solvers, and solved more problems, than avoidant individuals when tested with food- and escape-motivated tasks [200]. Exploratory Carib grackles were also faster learners and more likely to innovate in a foraging-extraction task than avoidant individuals [201]. However, this relationship is not always clearly defined. For example, both bold and shy chacma baboons improved their solving of a food extraction problem after watching a demonstrator [170]. Similarly, bold meerkats that approached a puzzle box first were not always the first to solve it [27], and neophobia did not significantly influence problem-solving ability in Barbary macaques *Macaca sylvanus* presented with puzzle boxes [202]. Although relationships between personality, behavioural flexibility and problem solving are not clearly defined, such individual variation should be taken into consideration when investigating developmental effects on problem solving.

## 4. Forgotten Components Limiting Our Understanding of Problem Solving and Its Development

Problem solving has been considered to rely almost exclusively on complex cognitive processes involving insightful thinking (i.e., just knowing what to do, rather than arriving at it through trial and error learning [181,203]), understanding of functionality [204] or causal understanding (i.e., being able to understand rules and consequences of actions [27]). Consequently, complex problem solving is often considered to be a consequence of relative brain size (e.g., birds and primates [169]). However, there is little evidence that problem solving involves complicated cognitive processes [28]. For example, introduced black rats *R. rattus* in Australia have caused extensive damage to macadamia *Macadamia* sp. orchards [205]. As rodents are evolutionarily constrained to gnaw due to the unrooted nature of their incisors [206], gnawing is an effective strategy for accessing novel food resources behind barriers or hard seed coats. To solve the problem of accessing the new food, black rats required only persistence, motivation and the appropriate mechanical apparatus rather than complex cognitive abilities. While each animal’s brain consists of a set of information-processing circuits that have evolved by natural selection to solve particular problems in their environment and increase their reproductive fitness [207], without the appropriate mechanical apparatus, the animal cannot solve the problem [208]. The ability to solve particular problems may therefore be species-specific, and morphologically constrained, specifically involving mechanical problem solving, unless animals can overcome these mechanical shortcomings (e.g., by developing tool use [28]).

Although problem solving has been studied in a wide variety of taxa, studies of the development of problem solving specifically have largely been restricted to birds [43], laboratory rats and mice [73,82,209], dogs [44], and primates that have been housed in captivity [131]. This is largely due to difficulties associated with observing free-living individuals [210] and accounting for their previous experience [95]. Consequently, studies rarely follow problem solving abilities over the development of individuals, instead comparing problem-solving ability between different age cohorts [168]. Such studies have shed light on the effects of intrinsic factors on the development of problem solving, but fail to consider individual variation in development.

Furthermore, the majority of studies on problem solving concern social species. Both solitary and social species need to problem solve, but the social environment could possibly influence how individuals develop their problem solving abilities. For example, social individuals may use social learning to problem solve, whereas solitary individuals would require persistence and motivation to achieve trial-and-error learning, or would rely on innovation because they are most likely unable to rely on social demonstrators for assistance [122,170], at least after weaning. Current studies therefore provide a limited view of the relevance of social conditions on problem solving development.

Finally, while the influences of environmental quality on problem-solving ability are documented, they are not well understood [27,40]. Animals tend to innovate under harsh conditions in times of necessity [24], yet good environmental conditions benefit problem solving by promoting neuroendocrine development [120] and reducing stress [118]. The effects of the physical or social environment tend to be studied either through manipulation studies during early development, with subsequent tests occurring later on as adults in static environments [165] or via correlative studies, where individuals from different habitats are compared [26]. Similarly, studies have investigated the impact of social rank [132], social isolation [211], group size [121,136], and group composition [2,27] on problem solving, but the majority of these studies have not explored the underlying developmental processes. To our knowledge, only one longitudinal study has tracked an individual’s problem-solving ability in response to changing physical environments. Cole et al. [40] found that individual performances in free-living great tits were consistent across time (seasonal variation). How problem-solving ability changes in response to changing social environments, such as when a subordinate changes dominance rank, has rarely been studied.

## 5. An Individual-Centric Focus can be Beneficial

The ability to solve a problem relies on a combination of genetic and non-genetic factors [44], physiology [97], behavioural flexibility [95], general cognitive ability [27], personality [129] and mechanical ability [212]. In addition, age and experience further influence problem-solving ability. Aging results in natural neuroendocrine system changes [213], which further affect behaviour and cognition [163]. However, every individual develops along its own unique developmental trajectory within the phylogenetic constraints of the species, and the relative contribution of these intrinsic and extrinsic factors and their interactions are likely to vary considerably between individuals. Therefore, we cannot assume that individuals from the same environment [214], or even the same clutch/litter [215], will behave or respond to the environment in the same way. We only have to look at genetic clones (e.g., identical human twins displaying linguistic differences [216]) to realise the uniqueness of individual developmental trajectories. This considerable variation argues strongly for focusing on individuals, particularly as they develop, learn and experience new things over their lifetimes in the context of problem solving. Therefore, when investigating problem solving abilities in the future, it may be beneficial to consider individual variation as an important aspect of the data analyses, and not just rejected as statistical ‘white noise’ (see [40,46] for examples). Using this approach may enable future research to identify key predictors, or clusters of common predictors, of problem-solving ability.

## 6. Conclusions

Individuals experience developmental changes over the course of their lifetimes, which impact their problem-solving abilities. The external environment, including the physical and social environments, can affect the development of problem solving via its impact on underlying genetic, non-genetic and neuroendocrine mechanisms. Problem solving has a heritable component in some species, while complex neuroendocrine processes are also involved in the development of problem solving. However, untangling the influence of these different factors on the development of problem solving is challenging, given their interdependence and complexity. Our understanding of how problem solving develops would benefit from studies of solitary species, to allow for comparisons of general causal mechanisms, since solitary species cannot rely on social learning about problems, at least after weaning. Furthermore, because environments are not static, future studies should consider the effects of changing environmental conditions over the course of an individual’s lifetime on the development of problem solving. Importantly, investigating individual variation in problem-solving ability is necessary for a full understanding of the development of problem solving, which will allow us to assess the relative contributions of different developmental factors on this ability in different individuals. 

## Figures and Tables

**Figure 1 animals-11-00866-f001:**
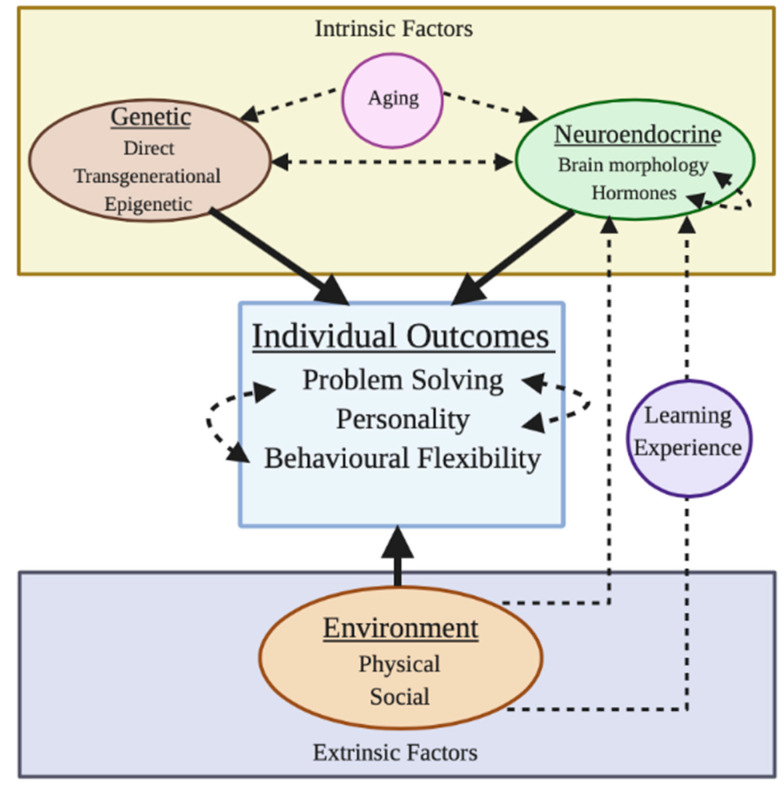
Intrinsic (genetic, neuroendocrine, and aging), extrinsic (environment) and acquired (learning and experience) factors influencing an individual’s development directly (solid arrows) or indirectly (dashed arrows). Arrow heads indicate direction of influence.

**Table 1 animals-11-00866-t001:** Definitions of problem solving and innovation quoted from the literature and associated references. We highlight the drivers (i.e., whether the ability to problem solve is linked to internal (e.g., physiology, cognition) or external (e.g., environmental) factors) and the properties of the animal (mechanical/morphological abilities or cognitive abilities) that authors attribute to problem solving.

Terminology	Drivers	Animal Properties	Definition	Reference
Innovation	Internal and External	Mechanical/Morphology and Cognitive	A new or modified learned behaviour not previously found in the population	[12]
Innovation	Internal and External	Mechanical/Morphology and Cognitive	The ability to invent new behaviours, or to use existing behaviours in new contexts A new or modified learned behaviour not previously found in the population A process that results in new or modified learned behaviour and that introduces novel behavioural variants into a population’s repertoire	[11]
Innovation	Internal and External	Mechanical/Morphology and Cognitive	The devising of new solutions	[13]
Innovation	Internal and External	Cognitive	An animal’s ability to apply previous knowledge to a novel problem or apply novel techniques to an old problem	[14]
Novel behaviour	Internal	Cognitive	The result of an orderly and dynamic competition among previously established behaviours, during which old behaviours blend or become interconnected in new ways	[15]
Physical problem solving	External	Mechanical/Morphology	Use of novel means to reach a goal when direct means are unavailable	[16]
Problem solving	Internal	Cognitive	Overcoming an obstacle that is preventing animals from achieving their goal immediately	[17]
Problem solving	External	Mechanical/Morphology and Cognitive	A problem exists when the goal that is sought is not directly attainable by the performance’ of a simple act available in the animal’s repertoire; the solution calls for either a novel action or a new integration of available actions	[18]
Problem solving	Internal	Cognitive	Any goal-directed sequence of cognitive operations	[19]
Problem solving	Internal and External	Mechanical/Morphology and Cognitive	A goal-directed sequence of cognitive and affective operations as well as behavioural responses for the purpose of adapting to internal or external demands or challenges	[20]
Problem solving	Internal	Cognitive	An analysis of means–end relationships	[21]
Problem solving	External	Mechanical/Morphology and Cognitive	A subset of instrumental responses that appear when an animal cannot achieve a goal using a direct action; the subject needs to perform a novel action or an innovative integration of available responses in order to solve the problem	[22]
Problem solving	Internal	Mechanical/Morphology	The ability to overcome obstacles and achieve a goal	[23]

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
