# Peer review of "Problem Solving in Animals: Proposal for an Ontogenetic Perspective"

_animals, 2021, doi:10.3390/ani11030866_

Round 1
Reviewer 1 Report
General
Wonderful review of a very essential topic for future animal welfare research (ie., mental state assessment and improvement). The contribution explores an interesting topic and make a very important suggestion for future research. Although it is not a systematic review of the scientific literature, it would be valuable to explicit the inclusion criteria of the review (line 59 and on). What keywords were required, and what search engines or scientific databases were explored? How did the authors justify excluding some literature in this review, and highlighting other literature?
Specific
Line 49-63: Here, and throughout the paper, it would be nice for the authors to distinguish experimental/artificial problem solving (ie., human created puzzle boxes, etc.), from natural/organic problem solving. What are the differences in species-specific goals, mechanisms, muscle motor patterns, etc. in solving a problem? Are all problems solved in the literature reviews related to accessing feed? (a more general audience will not understand the difference between the detour test, Lashley II maze, etc.)
Line 102: How many papers on the development of problem solving skills in non-human animals did the authors find in the literature search? While this statement of ‘paucity’ is find for a general introduction of a research paper, this is a review paper and should include a general quantity of literature published on this topic (for the audience to have perspective).
Line 137: Replace “&” with “and”
Line 141-142: If possible, it would be great if the authors could mention the task given to these different species for a problem solving study. Again, it provides much more context for the audience.
Line 184-190: Does the CA1 region of the hippocampus apply to all animals, or just mammals?
Lines 407-416: Could the authors incorporate the differences in play rate among juvenile and adult animals, and, the benefits of engaging in play behavior (practice skills, gain new skills, etc.).? Could this be a speculation as to the differences in problem solving success rate?
Reviewer 2 Report
The authors present a comprehensive review on the ontogenetic factors that may impact animals‘ problem solving abilities and outline why an individual-centric approach can help in elucidating the development of problem-solving.
Overall, I think the paper is well written and will make a valuable contribution to the topic as it adds important considerations to the body of literature on problem solving. However, there are a few aspects of the paper that I would like to see clarified/improved:
My main concern is that the introduction part needs a more careful literature review. In the current version of the mansucript single studies are reported where a review would be more appropriate.
For instance, ll.44-48 reporting on problem-solving reported in different taxa would beneft from citing a more comprehensive reviews available for those taxa.
For birds:
Lambert, M. L., Jacobs, I., Osvath, M., & von Bayern, A. M. (2019). Birds of a feather? Parrot and corvid cognition compared. Behaviour, 156(5-8), 505-594.
&
Taylor, A. H. (2014). Corvid cognition. Wiley Interdisciplinary Reviews: Cognitive Science, 5(3), 361-372.
For reptiles: Szabo, B., Noble, D. W., & Whiting, M. J. (2020). Learning in non‐avian reptiles 40 years on: advances and promising new directions. Biological reviews.
For fish: Kotrschal, K., Van Staaden, M. J., & Huber, R. (1998). Fish brains: evolution and anvironmental relationships. Reviews in Fish Biology and Fisheries, 8(4), 373-408.
For insects: Menzel, R., & Giurfa, M. (2006). Dimensions of cognition in an insect, the honeybee. Behavioral and Cognitive Neuroscience Reviews, 5(1), 24-40.
Furthermore, throughout the entire paper it is not really clear what the authors actually mean by „individual-centric approach“ when they suggest „focussing on individual animal“… As far as I understand, it means considering the individual variation in a population. However, the authors shall explain it more detailed and give examples.
Minor comments:
l.42 I think a more obvious example is reaching a fruit/branch that is out of reach (not necessarly related to the ability to climb
l.74-75 A good reference supporting this statement would McLean et al 2012 „How does cognition evolve?“
l.134-135 I would add here the hiritability of the g factor found in chimps Hopkins, W. D., Russell, J. L., & Schaeffer, J. (2014). Chimpanzee intelligence is heritable. Current Biology, 24(14), 1649-1652.
l. 414-416 A better performance found in juvenile subjects might be a result of higher playfullness (and exploration) at younger, whereas adult individuals might perform better because of increased experience rather than age-related changes on a neuroendocrine level. The authors discuss the effect of experience in the following paragraph, but I think it is important to mention the playfullness vs experience here.
l.551 While I fully agree that the individual variation should not longer be considered as white noise in the data, and developmental histories should be taken into account when evaluating problem-solving capacities of a species, I am confused what exactly the authors propose here. Again, it would help a reader if the authors would give a concrete example. Otherwise, one gets an impression that the authors propose that evidence from a single individual whose developmental history is known by observing and retesting it throughout its life is sufficient to demonstrate the cognitive capacity of a species.
Reviewer 3 Report
In this review, the authors deal with the problem-solving studies in animals, focused the literature review on the possible factors involved in the resolution of the problems. They stated that the variability in problem-solving abilities could arise due to differences in development, and other intrinsic (genetic, neuroendocrine and aging) and extrinsic (environmental) factors. At the end of their analysis, the authors suggest using an individual-centric approach in future research to better understand the development of problem-solving.
General comments:
The paper is well written, the writing is smooth, simple, and very readable. Overall, the review is interesting and could be useful for researchers working in this field.
It may be useful for the reader to have information on how the literature search was conducted so that he can repeat it. What keywords did the authors use to find relevant works? What databases or tools were used for the literature search (e.g. Scopus, scholar, Pubmed)?
Specific comments:
Table 1 legend: the first parenthesis has not been closed.
Reviewer 4 Report
I think this manuscript tackles a very relevant topic, it is a comprehensive and well organized review of the possible factors that influence problem solving in animals from a developmental perspective.
I only have some minor comments for the authors (listed below), which I hope could improve their final version:
line 21: I am not sure "complicated" is the most suitable adjective here. It is true that cognitive processes such as insight are not as easy to operationalize as associative learning, but that does not imply they are more complex (as the authors correctly point out elsewhere in their manuscript).
In the introduction I think the reader may require some clarification on the difference between behavioural flexibility and problem solving.
point 2.1.1 from line 131: There is an interesting paper on the relationship between cognitive skills in laying hens and their productivity
https://www.frontiersin.org/articles/10.3389/fpsyg.2018.02000/full
I think this paper would make a nice addition here in the section on direct genetic effects.
line 214 prolonged instead of prologed
line 220: high persistence is though also considered a sign of poor problem solving performance (see the literature on inhibitory control in animals).
line 557 which impact (instead of impacts)
Round 2
Reviewer 2 Report
The authors have dealt with all comments raised in the first review. I thank the authors for providing a very careful revision and a very detailed point-by-point reply. I am fully satisfied with the revised version and have no more points to raise.
Author Response
Thank you for your comments on the manuscript.